# Predicting the likelihood and intensity of mosquito infection from sex specific *Plasmodium falciparum* gametocyte density

John Bradley[1], Will Stone[2,3], Dari F Da[4], Isabelle Morlais[5], Alassane Dicko[6], Anna Cohuet[5], Wamdaogo M Guelbeogo[6], Almahamoudou Mahamar[5], Sandrine Nsango[7], Harouna M Soumaré[5], Halimatou Diawara[5], Kjerstin Lanke[2], Wouter Graumans[2], Rianne Siebelink-Stoter[2], Marga van de Vegte-Bolmer[2], Ingrid Chen[8], Alfred Tiono[9], Bronner Pamplona Gonçalves[3], Roland Gosling[8], Robert W Sauerwein[2], Chris Drakeley[3], Thomas S Churcher[10†*], Teun Bousema[2,3†]

[1]MRC Tropical Epidemiology Group, London School of Hygiene and Tropical Medicine, London, United Kingdom; [2]Radboud Institute for Health Sciences, Radboud University Medical Center, Nijmegen, Netherlands; [3]Department of Immunology and Infection, London School of Hygiene and Tropical Medicine, London, United Kingdom; [4]Institut de Recherche en Sciences de la Santé, Direction, Bobo Dioulasso, Burkina Faso; [5]Institut de recherche pour le développement, MIVEGEC (UM-CNRS 5290-IRD 224), Montpellier, France; [6]Malaria Research and Training Centre, Faculty of Pharmacy and Faculty of Medicine and Dentistry, University of Science, Techniques and Technologies of Bamako, Bamako, Mali; [7]Faculté de Médecine et des Sciences Pharmaceutiques, Université de Douala, Douala, Cameroon; [8]Global Health Group, Malaria Elimination Initiative, University of California, San Francisco, San Francisco, United States; [9]Centre National de Recherche et de Formation sur le Paludisme, Ouagadougou, Burkina Faso; [10]MRC Centre for Global Infectious Disease Analysis, Imperial College London, London, United Kingdom

*For correspondence:
thomas.churcher@imperial.ac.uk

†These authors contributed equally to this work

Competing interests: The authors declare that no competing interests exist.

**Abstract** Understanding the importance of gametocyte density on human-to-mosquito transmission is of immediate relevance to malaria control. Previous work (Churcher et al., 2013) indicated a complex relationship between gametocyte density and mosquito infection. Here we use data from 148 feeding experiments on naturally infected gametocyte carriers to show that the relationship is much simpler and depends on both female and male parasite density. The proportion of mosquitoes infected is primarily determined by the density of female gametocytes though transmission from low gametocyte densities may be impeded by a lack of male parasites. Improved precision of gametocyte quantification simplifies the shape of the relationship with infection increasing rapidly before plateauing at higher densities. The mean number of oocysts per mosquito rises quickly with gametocyte density but continues to increase across densities examined. The work highlights the importance of measuring both female and male gametocyte density when estimating the human reservoir of infection.
DOI: https://doi.org/10.7554/eLife.34463.001

## Introduction

Mosquitoes must ingest both male and female mature gametocytes to become infected with malaria. The shape of the relationship between gametocyte density and the probability of mosquito infection is thought to be complex for *Plasmodium falciparum* (*Churcher et al., 2013*; *Lin et al., 2016*; *Da et al., 2015*); but recent evidence has shown that the most comprehensive characterisation of this relationship was conducted using molecular tools that only quantified female gametocyte specific Pfs25 mRNA (*Churcher et al., 2013*; *Lasonder et al., 2016*). An additional shortcoming of these previous estimates is that female gametocytes were quantified from trendlines with unknown gametocyte sex-ratios, potentially affecting assay precision.

It is intuitive that gametocyte sex ratio is important in determining transmission efficiency (*Reece et al., 2008*; *Paul et al., 2002*). Gametocyte sex-ratio in *P. falciparum* is typically female biased but may differ between infections (*West et al., 2001*; *Robert et al., 2003*) and during infections (*West et al., 2001*; *Paul et al., 2000*), and may be influenced by immune responses that differentially affect male and female gametocytes (*Ramiro et al., 2011*). Fertility assurance explains why under certain conditions, such as a low gametocyte density, gametocyte sex-ratio may become less female biased to ensure all female gametes are fertilised (*Gardner et al., 2003*). Despite evidence that gametocyte sex ratio is adjusted in response to malaria developmental bottlenecks (*Reece et al., 2008*; *Paul et al., 2002*; *Neal and Taylor, 2014*), there is no direct evidence for its epidemiological importance. Since mosquito infections may occur from blood with *P. falciparum* gametocyte densities below the microscopic threshold for detection (*Churcher et al., 2013*; *Gonçalves et al., 2017*) and gametocyte density itself is an important determinant of sex ratio (*Reece et al., 2008*; *Robert et al., 2003*), sensitive quantification of male and female gametocytes is essential for assessments of the role of gametocyte sex ratio in natural infections and for reliable estimates of total gametocyte density. Here, we use a new molecular target of male *P. falciparum* gametocytes (Pf3D7_1469900 or PfMGET) (*Stone et al., 2017*) to explore the association between the density of female and male gametocytes and mosquito infection prevalence. The relationship between gametocyte density and oocyst density is also investigated as this may show a different relationship with gametocyte density (*Da et al., 2015*) and recent work has highlighted the potential importance of mosquito parasite density in mosquito-to-human transmission (*Churcher et al., 2017*). Both the number of infected mosquitoes and their parasite load may thus need to be considered when assessing the human reservoir of infection.

## Results

### Participant characteristics

Gametocyte carriers were included from Mali (n = 71), Burkina Faso (n = 64) and Cameroon (n = 13). *P. falciparum* parasite prevalence by microscopy ranged from ~40–70% between sites (*Gonçalves et al., 2017*; *Hien et al., 2017*; *Mahamar et al., 2017*; *Sandeu et al., 2017*); sampling occurred in different seasons (*Table 1*). All sites recruited gametocyte carriers prior to antimalarial treatment but used different enrolment criteria. Microscopically detected gametocyte carriers were included in three studies while one site (Balonghin) included submicroscopic gametocyte densities. Total gametocyte densities ranged from 0.04 to 1164 gametocytes/µL; the median proportion of male gametocytes was 25% (interquartile range 13–39%)(*Table 1*, *Figure 1*). Gametocyte density estimates by microscopy and qRT-PCR were positively correlated in all sites, but there was variation in the strength of correlation: Ouelessebougou (r = 0.74), Bobo Dioulasso (r = 0.27), Balonghin (r = 0.56) and Yaoundé (r = 0.91) (*Figure 1—figure supplement 1*). Across 148 successful membrane feeding experiments, 16.7% (1297/7757) of mosquitoes became infected, with considerable variation between gametocyte donors and study sites.

### Gametocyte sex ratios in natural infections

Female gametocyte densities quantified by Pfs25 quantitative reverse-transcriptase PCR (qRT-PCR) and male gametocyte densities by PfMGET qRT-PCR were positively correlated (*Figure 1A*;

**Table 1.** Characteristics of gametocyte carriers and mosquito feeding assays.

| | Ouelessebougou, Mali | Bobo Dioulasso, Burkina Faso | Balonghin, Burkina Faso | Yaoundé, Cameroon |
|---|---|---|---|---|
| Number of experiments | 71 | 19 | 45 | 13 |
| Enrolment criteria: | Detection of gametocytes by microscopy | Detection of gametocytes by microscopy | Detection of gametocytes by molecular QT-NASBA | Detection of gametocytes by microscopy |
| Period and season of data collection | January 2013-November 2014 (dry and wet season) | April-June 2016 (dry season) | October-November 2014 (wet season) | October-December 2015 (wet season) |
| *P. falciparum* parasite prevalence in cross-sectional surveys in the study area (microscopy) | 70.2% in children < 5 years, 2015–16 (*Mahamar et al., 2017*) | 40.9–61.7% in children 1–9 years, 2014–2015 (*Hien et al., 2017*) | 59.7% in children < 15 years, 2014 wet season (*Gonçalves et al., 2017*) | 44.7–55.6% in children 4–15 years, 2013–2014 (*Sandeu et al., 2017*) |
| Age, median (IRQ) | 11 (7–25) | 5–15 (range)* | 10 (8–13) | 9 (6–11) |
| Asexual parasite prevalence % (n/N) | 64.8 (46/71) | 73.7 (14/19) | 73.3 (33/45) | 76.9 (10/13) |
| Asexual parasite density per µL, median (IQR) | 432 (96–2880) | 360 (240–1040) | 658 (336–1237) | 944 (288–4224) |
| Total gametocyte density per µL, median (IQR) | 62.8 (31.4–146.8) | 19.2 (10.5–26.1) | 4.0 (0.6–11.0) | 64.4 (11.7–126.2) |
| Percentage of male gametocytes, median (IQR) | 14% (7–25%) | 51% (39–66%) | 30% (18–40%) | 32% (27–53%) |
| Number of mosquitoes examined per experiment, median (IQR) | 70 (63–79) | 29 (28–30) | 40 (35–45) | 37 (32–45) |
| Infectious individuals, % (n/N) | 74.7 (53/71) | 84.2 (16/19) | 22.2 (10/45) | 76.9 (10/13) |
| Infected mosquitoes, % (n/N) | 17.0 (842/4960) | 39.2 (208/531) | 3.5 (63/1783) | 38.1 (184/483) |

*the age of individual gametocyte donors was not recorded in Bobo Dioulasso; gametocyte carriers were recruited from the age range 5–15 years; asexual parasite density was determined by microscopy, gametocyte density by quantitative reverse transcriptase PCR. QT-NASBA = Pfs25 mRNA quantitative nucleic acid sequence based amplification.

DOI: https://doi.org/10.7554/eLife.34463.006

Spearman's rank correlation coefficient = 0.79, p<0.001). The fact that the relationship is linear on the log scale means that the percentage of gametocytes that were male decreased with increasing total gametocyte densities (*Figure 1B*, p<0.001 [Kruskal Wallis test]). This association remained apparent and statistically significant if any of the study sites with a smaller number of observations was removed (i.e. with removal of data from Bobo Dioulasso, Balonghin or Yaoundé). Conversely there was no evidence for an association between proportion of male gametocytes with asexual parasite density (*Figure 1C*, p=0.713).

## Infectivity in relation to gametocyte density

The proportion of mosquitoes developing oocysts is best described by a model that incorporates both the density of female and the density of male gametocytes (deviance information criterion, DIC = 451.5). In the best fit model, female gametocyte density explains most of the variability with the proportion of mosquitoes infected increasing rapidly with increasing gametocytaemia before saturating at high female gametocyte densities (*Figure 2A*). At female gametocyte densities of 200 per µl approximately 30% of mosquitoes are infected. At low gametocyte densities transmission appears to be impeded by a lack of male parasites (*Figure 2B*). The number of mosquitoes infected is on average lower for hosts with fewer than 50 male gametocytes per µl of blood, with the model predicting that male densities < 10 per µl reduces the proportion of infected mosquitoes by 50% (*Figure 2C*). Predictions for the proportion of mosquitoes infected according to the density of female and male gametocytes in the blood is given in *Figure 2D*. Poorer statistical fits were observed for models where the proportion of mosquitoes with oocysts was described by either the density of females alone (DIC = 481.3), the density of males (DIC = 538.8), or total gametocyte density (sum of male and female gametocytes, DIC = 501.7). There were considerable differences in the relative infectivity between sites. Compared to the Mali data, infectivity was similar in the study in Balonghin, Burkina Faso (0.95 times as high; 95% CI 0.1–1.4) but 3.65 times higher in Bobo

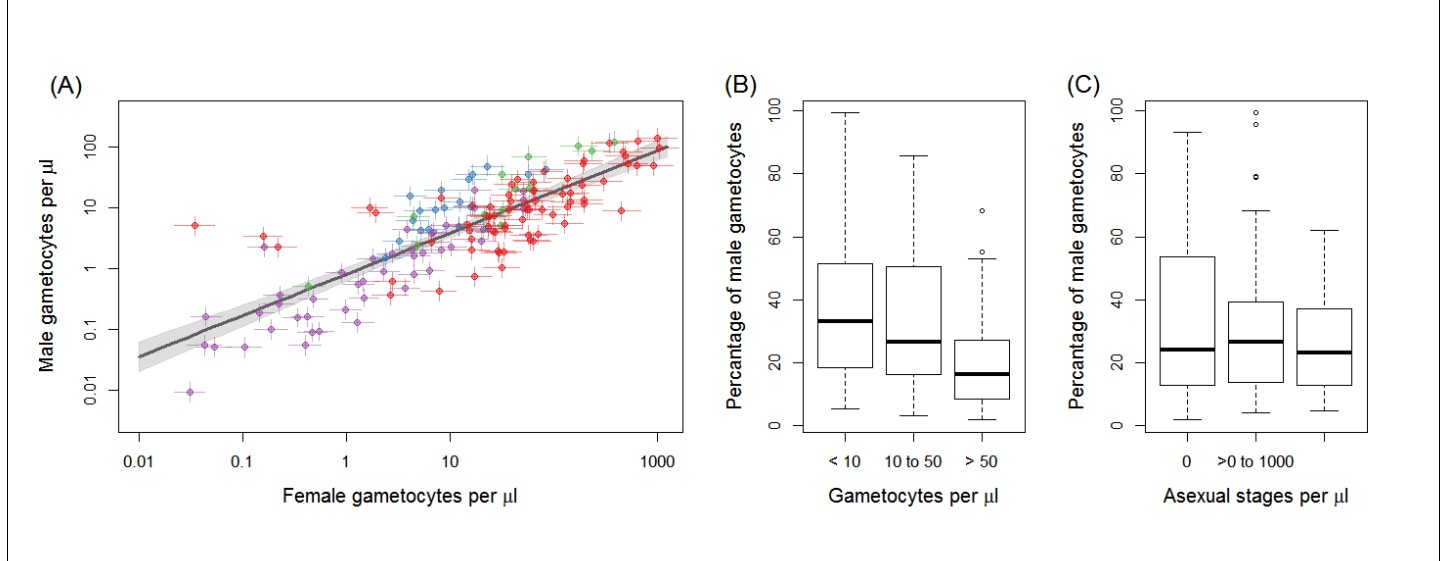

**Figure 1.** Gametocyte density in natural infections. The density of female gametocytes and male gametocytes is presented in panel (**A**) for samples from Ouelessebougou, Mali (red), Yaoundé, Cameroon (green), Bobo-Dioulasso, Burkina Faso (blue) and Balonghin, Burkina Faso (purple). Female and male gametocyte densities were positively associated (r = 0.79, p<0.001) with the best fit relationship shown by the black solid line (grey shaded area showing 95% the confidence interval around this line). Coloured horizontal and vertical lines indicate Bayesian credible intervals (CIs) around point estimates. The proportion of gametocytes that were male was negatively associated with total gametocyte density (**B**) but not with asexual parasite density (**C**). All raw data can be found in *Figure 1—source data 1* whilst a description of the relationship modelled in (**A**) is provided in *Figure 1—source data 2*. The relationship between gametocyte density as measured by microscopy and PCR is given for each site in *Figure 1—figure supplement 1*.

DOI: https://doi.org/10.7554/eLife.34463.002

The following source data and figure supplement are available for figure 1:

**Source data 1.** Raw data presented in *Figure 1*.

DOI: https://doi.org/10.7554/eLife.34463.004

**Source data 2.** Description of the statistical model determining the shape of the relationship between female gametocyte density and male gametocyte density.

DOI: https://doi.org/10.7554/eLife.34463.005

**Figure supplement 1.** Relationship between total gametocyte densities as measured by microscopy or female gametocyte densities quantified by Pfs25 quantitative reverse-transcriptase PCR.

DOI: https://doi.org/10.7554/eLife.34463.003

Dioulasso, Burkina Faso (95% CI 2.3–5.2) and 1.68 times higher in site in Yaoundé, Cameroon (95% CI 1.1–2.6) (*Figure 2—figure supplement 1*, *Figure 2—source data 2*.). Other models, including both qRT-PCR and microscopy, showed poorer performance in predicting mosquito infection prevalence (*Figure 2—source data 2*).

## Oocyst density in mosquitoes

The distribution of oocysts between mosquitoes is highly over-dispersed, with some mosquitoes harbouring very high oocyst densities. This aggregated distribution is reflected in the relationship between proportion of mosquitoes infected and mean oocyst density (*Figure 3A*). Mean oocyst density increased with increasing gametocyte density and continued to increase across the range of gametocyte densities observed, without evidence for a plateau being reached (*Figure 3B*). Total gametocyte density (sum of females and males) was able to predict mean oocyst density better than female gametocyte density alone. The higher complexity of the density model (requiring the fitting of the negative binomial distribution) meant that it was not possible to fit a model analogous to the best fit model on infectivity (i.e. a function dependent on female gametocyte density but with reduced transmission at low male densities). Care should be taken interpreting the relative importance of females and male gametocytes in predicting oocyst density as this result should be validated with a larger dataset.

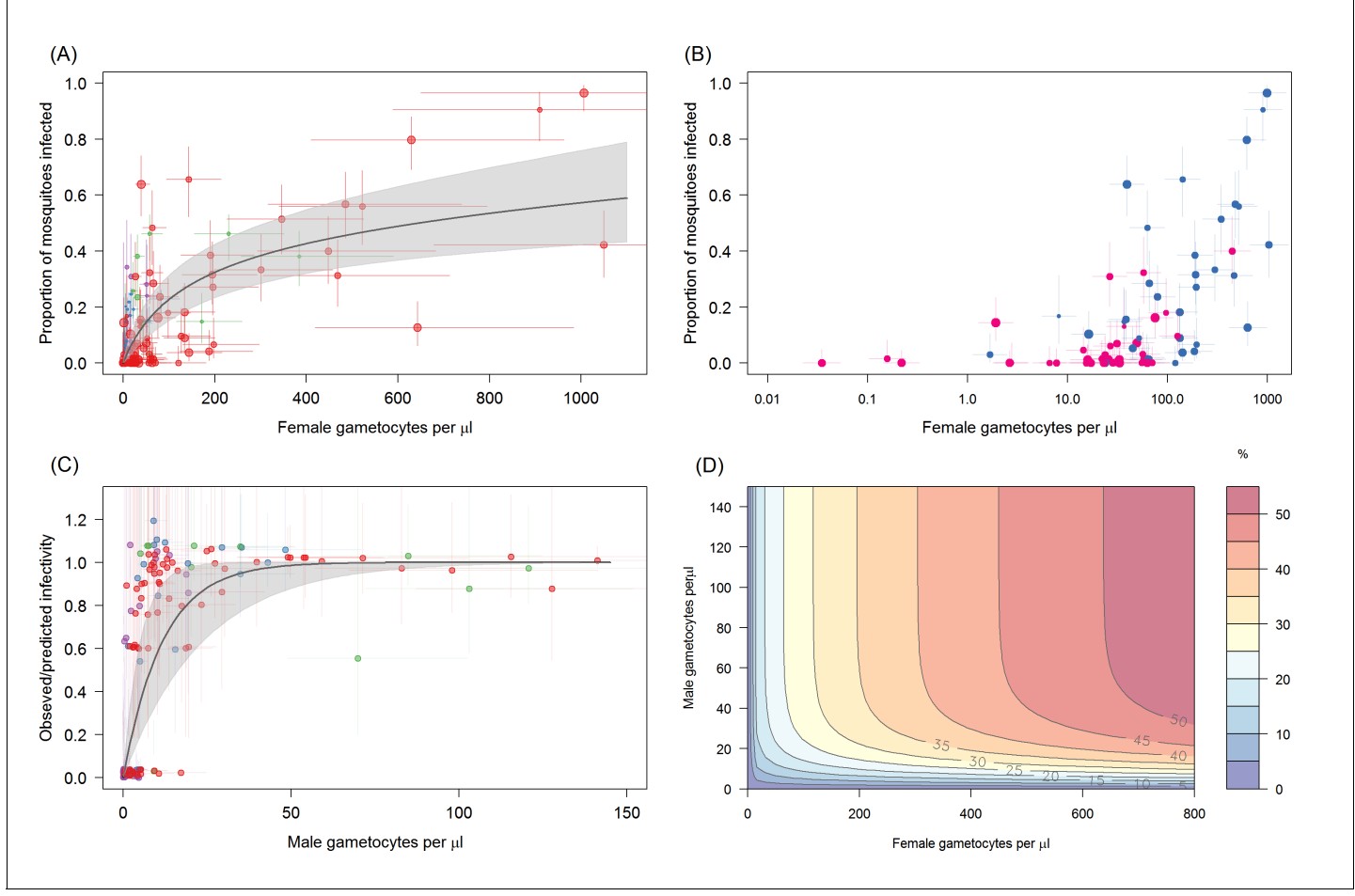

**Figure 2.** The relationship between *Plasmodium falciparum* gametocyte density and the proportion of mosquitoes that develop oocysts. (A) The association with female gametocyte density. The solid black line indicates the best-fit statistical model with grey shaded 95% Bayesian credible intervals (CI). Infectivity depends on both the density of male and female gametocytes so the figure uses the relationship between male and female gametocyte density defined in *Figure 1A* to predict overall transmission. Point colour denotes the study from which the observation came (red = Ouelessebougou, Mali, green = Yaoundé, Cameroon, blue = Bobo Dioulasso, Burkina Faso and purple = Balonghin, Burkina Faso) and point size is proportional to the number of mosquitoes dissected. Horizontal and vertical lines indicate 95% CIs around point estimates. To aid clarity the figure shows points and model predictions scaled to the largest dataset (i.e. each site was scaled by the relative infectivity compared to the Mali dataset). A version of the figure without this scaling is shown in *Figure 2—figure supplement 1*, which shows all raw data and separate model predictions for each site. (B) Relationship between female gametocyte density and the proportion of infected mosquitoes for different male gametocyte densities for experiments from Mali (n = 71). Points are coloured according to the density of male gametocytes (<10 male gametocytes/μL = pink,≥10 male gametocytes/μL = dark blue). Note bloodmeals containing lower numbers of male parasites typically have lower infectivity for a given female density. *Figure 2—figure supplement 2* shows the same figure but differentiating between points using the sex ratio instead of absolute male density. (C) Model predictions for the reduction in the proportion of mosquitoes infected due to male gametocyte density. Data points are the observed infectivity divided by the predicted infectivity as predicted by the statistical model using the density of female gametocytes in the sample (colours matching panel A). Values less than one indicate reductions in relative transmission. The solid black line shows the best fit model for this restriction from the model in 2A, with shaded area and horizontal and vertical lines indicating 95% CIs. (D) illustrates the best fit model predictions for the 3D relationship between female gametocyte density, male gametocyte density and the percentage of mosquitoes which develop oocysts (colour scale from 0 to ≥50% infected mosquitoes, see legend). All raw data can be found in *Figure 2—source data 1* whilst statistical comparisons of the different curves tested in (A) are provided in *Figure 2—source data 2*.

DOI: https://doi.org/10.7554/eLife.34463.007

The following source data and figure supplements are available for figure 2:

**Source data 1.** Raw data presented in *Figure 2*.
DOI: https://doi.org/10.7554/eLife.34463.010
**Source data 2.** Description of the statistical model determining the shape of the relationship between gametocyte density and mosquito infection.
DOI: https://doi.org/10.7554/eLife.34463.011
*Figure 2 continued on next page*

*Figure 2 continued*

**Figure supplement 1.** Site-specific differences in the relationship between *Plasmodium falciparum* female gametocyte density and the proportion of mosquitoes that develop oocysts.

DOI: https://doi.org/10.7554/eLife.34463.008

**Figure supplement 2.** Relationship between female gametocyte density and the proportion of infected mosquitoes for different male gametocyte densities for experiments from Mali (n = 71).

DOI: https://doi.org/10.7554/eLife.34463.009

## Discussion

We present an improved model to predict mosquito infection from female and male gametocyte density estimates. Previous molecular assessments of *P. falciparum* gametocyte density quantified female specific Pfs25 mRNA then converted this into a measure of gametocytes per μL of blood by reference to standard curve of mixed-sex gametocytes (*Churcher et al., 2013*; *Wampfler et al., 2013*; *Ouédraogo et al., 2016*). These assessments therefore quantified neither female nor total gametocyte density accurately. The current model presents a considerable improvement over previous work (*Churcher et al., 2013*), both by separately quantifying male and female gametocytes using sex-specific mRNA markers and standard curves (*Lasonder et al., 2016*) and also by using considerably more accurate estimates of total gametocyte density. Male and female gametocytes were quantified separately using automated extraction of nucleic acids (*Stone et al., 2017*). Manual extraction can result in considerable variation (*Walker et al., 2015*) that may have affected the accuracy of our earlier gametocyte estimates (*Churcher et al., 2013*). In addition, we used qRT-PCR that

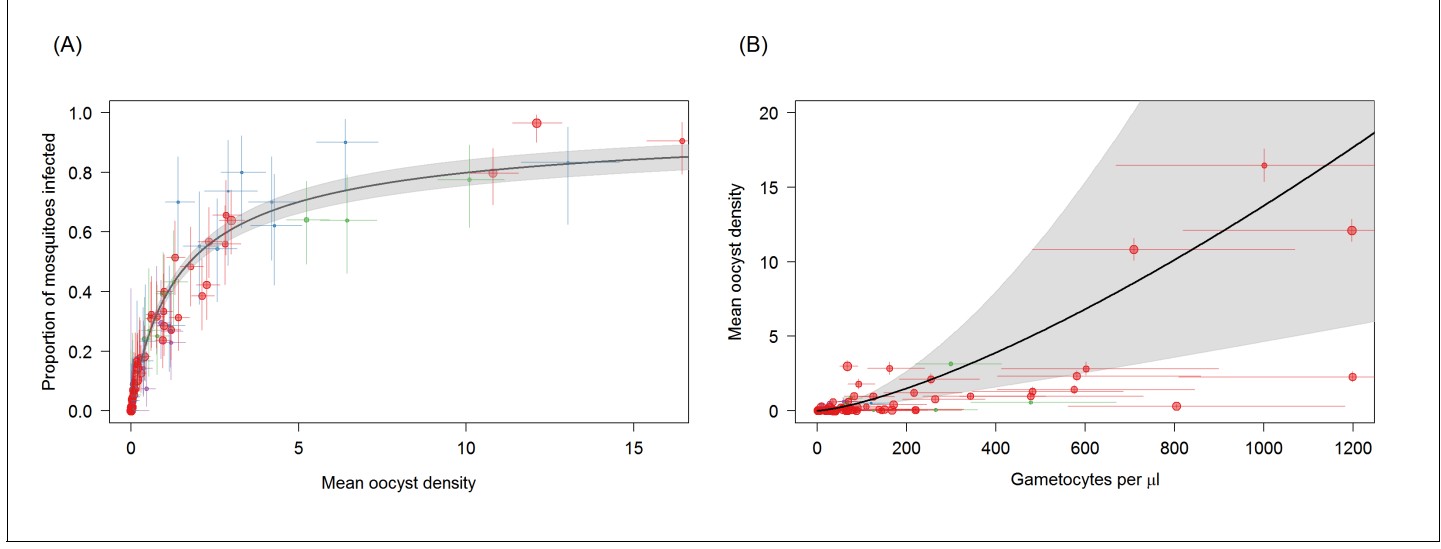

**Figure 3.** Associations between mean oocyst density; the proportion of mosquitoes that develop oocysts; and gametocyte density. (**A**) The relationship between mean oocyst density and the proportion of mosquitoes that develop oocysts (red = Ouelessebougou, Mali, green = Yaoundé, Cameroon, blue = Bobo Dioulasso, Burkina Faso, purple = Balonghin, Burkina Faso; point size is proportional to the number of mosquitoes dissected). A Hill function gave the best fit to these data (DIC linear = 1310; power = 602; Hill = 434). (**B**) The relationship between total gametocyte density and the mean oocyst density in all mosquitoes. The power function gave the best fit to these data (DIC linear = 1042; hyperbolic = 1044; gompertz = 1067; power = 1036); total gametocyte density gave a better fit than only female gametocyte density (best DIC = 1061). Horizontal and vertical lines indicate 95% Bayesian credible intervals (CIs) around point estimates, solid black line indicates the best-fit model with grey shaded area indicates the 95% CI around this line. Like *Figure 2A*, Panel 3B aids clarity by scaling model and data to the average infectivity of the largest dataset though the raw data and model fits are provided in *Figure 3—figure supplement 1*.

DOI: https://doi.org/10.7554/eLife.34463.012

The following figure supplement is available for figure 3:

**Figure supplement 1.** Site-specific differences in the relationship between *Plasmodium falciparum* female gametocyte density and the proportion of mosquitoes that develop oocysts.

DOI: https://doi.org/10.7554/eLife.34463.013

has higher precision than the previously used quantitative nucleic acid sequence based amplification, QT-NASBA (*Pett et al., 2016*).

The analyses show that the relationship between the density of gametocytes and transmission can be adequately described by a simple saturating relationship, and that a more complicated curve with two inflection points (which generates a double plateau) does not improve model fit (*Churcher et al., 2013*). The change in shape of the best fit relationship has implications for predicting the effect of transmission blocking interventions such as gametocytocidal drugs. The first plateau of the previous relationship (*Churcher et al., 2013*) predicted similar infectivity for gametocyte densities between 1 and 200 gametocytes per µl. If this was the case then an intervention that killed 99% of gametocytes would have very little effect on infectivity on a person with a density of 200 gametocytes per µl but impacted transmission considerably below this level. The new relationship presented here suggests an intervention reducing gametocyte density would have an impact on transmission to mosquitoes across the entire range of gametocyte densities observed in the field with, for example, a predicted reduction in mosquito infection prevalence of 32% to 1% when female gametocyte density is reduced from 200 to 1 female gametocytes per µl. The new work indicates that even gametocytocidal drugs with incomplete efficacy would reduce transmission.

The increased simplicity found here compared to the previously described relationship is likely to be driven by the improved accuracy of gametocyte density estimates as models fit solely to female parasites density had a similar simplified shape. The concurrent quantification of female and male gametocytes leads to an improved estimation of total gametocyte biomass and could explain why models with information on both sexes were more accurate. Similarly, combining data from microscopy and PCR could also improve overall precision when both techniques have relatively high measurement error. Though improved accuracy of gametocyte density estimates through combining imprecise measures cannot be discounted the model fit to total gametocyte density (male plus female) gave a poorer fit than the model where transmission was dependent on female density unless there were very low male parasite densities. This suggests that the number of male gametocytes may become a limiting factor for transmission at low gametocyte densities, although future studies that directly relate infectivity to mosquitoes to female and male gametocyte densities are needed to confirm this association. Such studies may benefit from an assay that targets a transcript with identical expression levels in male and female gametocytes (*Meerstein-Kessel et al., 2018*) or a multiplex male-female gametocyte assay that avoids uncertainties in quantifying total gametocyte biomass from two separate assays.

The work confirmed previous evidence that the proportion of male gametocytes is negatively associated with total gametocyte density (*Reece et al., 2008*; *Robert et al., 2003*; *Sowunmi et al., 2008*), which may reflect a strategic investment in male gametocytes to maximize the likelihood of transmission in low-density infections (fertility assurance) (*Reece et al., 2008*; *Reece et al., 2009*). The models also indicated that male gametocyte density may become a limiting factor for transmission success at low gametocyte densities. These observations are in line with in vitro findings with *P. falciparum*, *Plasmodium berghei* and *Plasmodium chabaudi* where infections with a higher proportion of male gametocytes gave higher transmission success at low gametocyte densities but reduced success at higher gametocyte densities (*Reece et al., 2008*; *Mitri et al., 2009*).

The presented model considerably improved the prediction of mosquito infection rates compared to our previous manuscript (*Churcher et al., 2013*). Nevertheless, some level of uncertainty remains, particularly between study sites with several experiments resulting in considerably lower mosquito infection rates than predicted based on gametocyte density and sex-ratio. There are several plausible reasons for this. Naturally acquired antibody responses to gametocyte antigens may reduce transmission efficiency and form a first explanation why many mosquitoes may fail to become infected when feeding on some hosts with high-density gametocyte infections. Whilst immune responses that completely prevent mosquito infections are only sporadically detected in naturally exposed populations (*Stone et al., 2018*), it is plausible that functional transmission reducing immunity has reduced mosquito infection rates for a proportion of gametocyte carriers in our study. Temporal fluctuations in transmission reducing immunity (*Ouédraogo et al., 2011*) may also have contributed to the apparent differences in infectivity between study sites that recruited gametocyte carriers at different time-points in the season. The site with the highest infectivity in the current study (Bobo Dioulasso), recruited individuals in the dry season when the impact of transmission reducing immunity may be lowest (*Ouédraogo et al., 2011*). The extent of between site heterogeneity in

transmission should be interpreted with caution since two of the four sites had <20 study participants. At all sites malaria-infected individuals were recruited prior to malaria treatment, making it unlikely that drug-induced sterilization of circulating gametocytes may have affected our analyses. Variation in the susceptibility of the mosquito colonies to parasite genotypes may provide a second plausible reason for the imperfect model fit (*Lefèvre et al., 2013*). A third hypothesis could be that our quantification of gametocytes by Pfs25 and PfMGET qRT-PCR may not fully reflect gametocyte maturity and infectivity as gametocytes may be detectable in the blood stream before (*Hallett et al., 2006*) and after peak infectivity is reached (*Ouédraogo et al., 2016*). Lastly, technical issues related to RNA degradation may affect gametocyte quantification and thus yield unreliable low gametocyte density estimates (*Pett et al., 2016*). There were no apparent issues in maintaining the essential temperature control in the field, during transportation, or following RNA extraction for any of the samples included in this study, nor did the associations between microscopy and qRT-PCR gametocyte density estimates indicate (site-specific) RNA degradation that may have resulted in underestimations of true gametocyte densities.

The study shows that though the prevalence of mosquitoes with oocysts plateaus at high gametocyte densities the average number of oocysts in those mosquitoes continues to rise, as previously reported for *P. vivax* (*Kiattibutr et al., 2017*) and *P. falciparum* (*Da et al., 2015*). There were insufficient data to fit a model where oocyst density was dependent on female density but with a penalty for low male densities which was the best fit model for infectivity. Whether such a model would fit the oocyst density data better that the model presented here is a question for future research.

Understanding the association between gametocyte density and mosquito infection rates is of immediate relevance for malaria control efforts (*Lin et al., 2016*; *Slater et al., 2015*; *Gonçalves et al., 2017*). Here we show that accurate measures of female and male gametocyte density can better predict human-to-mosquito infection, and could be used to assess the infectiousness of human populations.

## Materials and methods

### Study populations and mosquito feeding experiments

Field samples were collected at four malaria endemic sites. Samples were collected prior to treatment and after written informed consent was obtained from participants or their guardian(s). Ethical clearance was provided by the National Ethics Committee of Cameroon; Ethical Review Committee of the Ministry of Health, Burkina Faso; Ethics Committee of the Malaria Research and Training Centre, Bamako; Ethics review committee Centre MURAZ; University of California, San Francisco, and London School of Hygiene and Tropical Medicine. Procedures for Ouelessebougou, Mali are described elsewhere (*Dicko et al., 2016*). From the trial in Ouelessebougou, baseline samples from microscopically detected gametocyte carriers were used. Additional samples were collected from asymptomatic microscopically detected gametocyte carriers aged 5–15 years in Bobo Dioulasso in Burkina Faso and Yaoundé, Cameroon. Lastly, a random selection of samples from a xenodiagnostic study in Balonghin in Burkina Faso was used (*Gonçalves et al., 2017*). Samples were eligible for selection if gametocytes were detected by Pfs25 QT-NASBA that has an estimated lower limit of detection between 0.02–0.1 gametocytes/μL and thus provided data-points at the lower range of gametocyte densities (*Gonçalves et al., 2017*). The same membrane feeding protocol was used at all sites: local *Anopheles coluzzii* colony mosquitoes (Mali, Cameroon and Bobo Dioulasso, Burkina Faso) or colony mosquitoes comprising a mixture of *A. coluzzii*, *A. gambiae s.s.* and hybrid forms (Balonghin, Burkina Faso) were allowed to feed for 15–20 min on heparin blood samples until dissection in 1% mercurochrome at day seven post-feeding and oocyst detection by two independent microscopists (*Ouédraogo, 2013*). For all sites membrane feeding and sample collection were performed prior to antimalarial treatment

### Molecular analysis of samples from naturally infected gametocyte donors

Female gametocytes were quantified by quantitative reverse transcriptase PCR (qRT-PCR) targeting female Pfs25 mRNA, as described elsewhere in detail (*Stone et al., 2017*) based on established protocols (*Wampfler et al., 2013*). For male gametocytes we used a recently developed qRT-PCR

(*Stone et al., 2017*) based on *PfMGET* (male gametocyte enriched transcript, *Pf3D7_1469900*), a transcript that is highly enriched in male *P. falciparum* gametocytes (*Lasonder et al., 2016*). Primer sequences are provided in *Table 2*. For all qRT-PCR, mRNA was extracted from blood collected in EDTA tubes by venipuncture; 100 µL of whole blood was stored at −80°C in 500 µL RNAprotect (Qiagen; for Burkina Faso and Cameroon samples) or 900 µL L6 buffer (Severn Biotech, Kidderminster, UK; for Mali samples) until automated extraction using a MagNAPure LC (Total Nucleic Acid Isolation Kit–High Performance; Roche Applied Science, Indianapolis, IN, USA). cDNA was synthesised directly from nucleic acids for the *PfMGET* assay, for which the primers are intron-spanning, and after DNase treatment (RQ1 DNase I Digest Kit, Promega) for the *Pfs25* assay, using High Capacity cDNA Reverse Transcription Kits (Applied Biosystems, Foster City, CA). qRT-PCR results were converted to male and female gametocyte densities using standard curves (ten-fold serial dilutions from $10^6$ to 10 gametocytes/ml) of separate male and female gametocyte populations that were generated using a transgenic parasite line expressing a male specific fluorescence marker (*Lasonder et al., 2016*; *Stone et al., 2017*). The purity of male and female trendlines was previously confirmed by staining of sorted gametocyte populations using female gametocyte specific anti-Pfg377 antibodies (*Stone et al., 2017*; *Suárez-Cortés et al., 2016*). For both Pfs25 and PfMGET qRT-PCR, a threshold for positivity was set at one gametocyte per sample (0.01/µL).

## Statistical analysis

The statistical methods used here are the same at those in the original paper (*Churcher et al., 2013*), which we briefly recapitulate here. qRT-PCR results are in the form of cycle-thresholds (CT, which is the number of cycles it takes for the fluorescence associated with target amplification to exceed a defined threshold. The relationship between CT and gametocyte density is estimated by fitting a linear regression to CT estimates generated using a sample with known gametocyte density (a 10-fold dilution series). Let the observed CT be denoted by y then,

$$y = \beta_0 + \beta_1 \ln x + \varepsilon \tag{1}$$

where $\beta_0$ and $\beta_1$ are regression coefficients estimates, x is the (known) parasite density from the dilution series and $\varepsilon$ represents a normally distributed random error ($\varepsilon \sim N(0, s^2)$). *Equation (1)* can be rearranged to enable us to estimate gametocyte density from a CT measurement. We use a Bayesian hierarchical model to estimate the coefficients $\beta_0$ and $\beta_1$.

These gametocyte density estimates are used to determine the relationship between gametocyte density and the proportion of mosquitoes developing oocysts. Four functional forms (linear, power, hyperbolic and Gompertz [*Churcher et al., 2013*]) were each fit four times: on female gametocyte density alone; on male gametocyte density alone; on total gametocyte density; and finally female gametocyte density but multiplied by a function accounting for reduced transmission at low male densities. Terms allowing for different infectivity at each site were incorporated into the models. The algebraic forms of these models are given in *Figure 2—source data 2*. The model quantifying the uncertainty in gametocyte density estimates was fit at the same time as the regression determining the relationship between gametocyte density and infectivity using Bayesian Markov Chain Monte Carlo methods assuming a Binomial error structure for each feeding experiment. Fitting the models simultaneously enabled the uncertainty in the gametocyte density estimates to be reflected in the uncertainty of the shape of the relationship. The models were compared using the DIC with a lower value indicating the most parsimonious fit.

The relationship between gametocyte density and oocyst density was examined in an analogous way, with two exceptions: (1) A negative binomial error structure was used to describe oocyst counts and (2) The increased complexity of the model precluded the inclusion of the function accounting

**Table 2.** Primer sequences for the Pfs25 female marker, and male marker PfMGET.

| Gene target | Forward primer | Reverse primer |
| --- | --- | --- |
| *Pfs25* | GAAATCCCGTTTCATACGCTTG | AGTTTTAACAGGATTGCTTGTATCTAA |
| *PFMGET* | CGGTCCAAATATAAAATCCTG | GTGTTTTTAATGCTGGAGCTG |

DOI: https://doi.org/10.7554/eLife.34463.014

for reduced transmission at low male densities, so each functional form was only fit twice; on female gametocyte density and total gametocyte density.

## Acknowledgements

This work was supported by the Bill and Melinda Gates Foundation (AFIRM OPP1034789 & INDIE OPP1173572). TB and WS are further supported by a fellowship from the European Research Council (ERC-2014-StG 639776). The PATH Malaria Vaccine Initiative funded the collection of samples in Bobo Dioulasso, Burkina Faso and Yaoundé, Cameroon.

## Additional information

### Funding

| Funder | Grant reference number | Author |
|---|---|---|
| Bill and Melinda Gates Foundation | AFIRM OPP1034789 & INDIE OPP1173572 | Chris Drakeley<br>Teun Bousema |
| H2020 Excellent Science | ERC-2014-StG 639776 | Will Stone<br>Teun Bousema |
| PATH | Malaria Vaccine Iniative | Dari F Da<br>Isabelle Morlais<br>Anna Cohuet<br>Teun Bousema |

The funders had no role in study design, data collection and interpretation, or the decision to submit the work for publication.

### Author contributions

John Bradley, Conceptualization, Formal analysis, Investigation, Visualization, Writing—original draft, Writing—review and editing; Will Stone, Conceptualization, Investigation, Methodology, Writing—original draft, Writing—review and editing; Dari F Da, Isabelle Morlais, Alassane Dicko, Anna Cohuet, Wamdaogo M Guelbeogo, Almahamoudou Mahamar, Sandrine Nsango, Halimatou Diawara, Kjerstin Lanke, Wouter Graumans, Rianne Siebelink-Stoter, Marga van de Vegte-Bolmer, Ingrid Chen, Alfred Tiono, Investigation, Methodology, Writing—review and editing; Harouna M Soumaré, Investigation, Methodology; Bronner Pamplona Gonçalves, Resources, Investigation, Methodology, Writing—review and editing; Roland Gosling, Robert W Sauerwein, Investigation, Writing—review and editing; Chris Drakeley, Conceptualization, Supervision, Funding acquisition, Investigation, Project administration, Writing—review and editing; Thomas S Churcher, Conceptualization, Formal analysis, Investigation, Visualization, Writing—original draft; Teun Bousema, Conceptualization, Data curation, Supervision, Funding acquisition, Investigation, Methodology, Writing—original draft, Project administration

### Author ORCIDs

John Bradley http://orcid.org/0000-0002-9449-4608
Wouter Graumans http://orcid.org/0000-0003-3952-6491
Chris Drakeley http://orcid.org/0000-0003-4863-075X
Thomas S Churcher http://orcid.org/0000-0002-8442-0525
Teun Bousema http://orcid.org/0000-0003-2666-094X

### Decision letter and Author response

Decision letter https://doi.org/10.7554/eLife.34463.019
Author response https://doi.org/10.7554/eLife.34463.020

## Additional files

### Supplementary files
• Transparent reporting form
DOI: https://doi.org/10.7554/eLife.34463.015

### Data availability
All raw data can be found in source data supplements of the relevant figures.

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
