## [Decision Letter]

[Editors’ note: a previous version of this study was rejected after peer review, but the authors submitted for reconsideration. The first decision letter after peer review is shown below.]

Thank you for submitting your work entitled "Predicting mosquito infection rates and infection intensity from *Plasmodium falciparum* gametocyte density and sex ratio" for consideration by *eLife*. Your article has been reviewed by two peer reviewers, and the evaluation has been overseen by a Reviewing Editor and a Senior Editor. The following individuals involved in review of your submission have agreed to reveal their identity: Edward A. Wenger (Reviewer #1).

Our decision has been reached after consultation between the reviewers. Based on these discussions and the individual reviews below, we regret to inform you that your work in the present format will not be considered further for publication in *eLife*. However, the authors are encouraged to make all the suggested revisions and submit the manuscript as a new submission for review.

This study builds on results from previously published *eLife* paper in that the authors now include new estimates of male gametocytes densities and that provides more detailed prediction of the human-to- mosquito infectiousness. What follows is a summary of the main issues and detailed comments from the reviewers are attached below.

1) The new laboratory methods (quantification of male gametocytes and greater precision for female gametocytes) are described in other publications, and only very briefly here. Please expand on this topic in the discussion.

2) Even with the final model including male gametocyte densities, there is uncertainty. There are substantial between-site differences, and within sites there is unexplained variation with more than a few points lying away from the fitted lines. Please provide explanations.

3) There are a number of issues with the data analysis (correct data, sample size in one site, highly influential points, investigation into the Balonghin data). Please, re-analyze the results and provide a significantly revised interpretation.

4) It is stated that there is a change in the shape of the relationship between gametocyte densities and infectivity by including male gametocyte densities. However, the model may have greater prediction accuracy since two methods to estimate parts of the gametocyte population are used rather than one (the male and female gametocyte densities are added together in the model). In addition, the shape of the relationship does not appear to differ with the current datasets if only female gametocytes are used, compared to male and female together (the best fit for both would be the power function). It may be that there is a difference between the previous data and the current datasets.

Reviewer #1:

The subject of human-to-mosquito infectiousness and its determination by gametocyte density and other factors is a very important one. The authors make an important contribution here that is of general interest.

I have three core concerns with the paper that should be addressed:

1) The paper claims that we can now "reliably predict" mosquito infectiousness by including male and female gametocyte density quantification. However, the significant site-specific differences and other sources of unexplained variance are barely discussed.

- Given the role that seasonality and immunity play on the underlying density distributions (Ouedraogo, 2015; Gerardin, 2015), more information in the Materials and methods section would be appreciated on year and season of sampling (e.g. January 2013 to November 2014 enrollment), enrollment protocol (e.g. microscopy positive gametocytes), and immunity (e.g. incidence per year) beyond "see Dicko, 2016" (not open access).

- You should discuss the implications of the Bobo, Cameroon, and Mali data consisting of microscopy-positive gametocyte carriers, but the Balonghin site only requiring Pfs25 positive.

- You should include this information in Table 1, so the reader isn't forced to discover for herself the reasons for the very different gametocyte density ranges in spite of more similar asexual density ranges.

- Why is infectivity 3.65 times higher in Bobo-Dioulasso? The list of possible differences put forth in the Discussion section, "vector permissiveness, local parasite strains, human [immunity]" is vague and only serves to propagate the general confusion in the community. Without a satisfactory understanding, the claim of "reliably predicting" infectiousness is not very strong.

2) This paper claims to make an important breakthrough in improved precision on female gametocyte density quantification. The Discussion section touches on this, but the implications of this are important and should be expanded upon.

- Presently, there is insufficient detail on the determination of confidence intervals on density quantification. The only hint is the Materials and methods section sentence that they are "based on plate-specific calibration trendlines".

- An example of where I doubt the veracity of the error quantification is in Figure 2B, where there are two samples with very low density, in spite of both being microscopically positive. Note the microscopy detection threshold is almost 2 orders of magnitude higher, or ~10σ if error bars are 95% CI. Were these samples extreme outliers in leukocyte counts and the microscopy threshold is an invalid comparison? Does the female gametocyte density measurement error have significantly wider non-Gaussian tails?

3) A core assertion in this paper is that the addition of male gametocyte densities improves prediction. This could be caused by a combination of the following two factors: (1) mechanistically there is a threshold below which there are not enough male gametocytes present in the blood meal; or (2) the combination of two independent imprecise density measures gives a more accurate estimate of the "true" total gametocyte densities and hence more predictable infectiousness.

The authors are strongly suggesting the first mechanism. However, there are indications that the latter mechanism is also playing an important role - take for example the Mali points that populate the lower-right quadrants of Figure 1A and 2A.

- A version of Figure 2B colored according to male gametocyte ratio would help discern how important each effect might be.

- A direct look at the Balonghin data, which in most plots is hiding in the corner of linear plots, would be instructive as it also includes submicroscopic gametocyte infections.

The source data has Balonghin and Bobo swapped compared to what appears in Table 1. The authors should verify this hasn't introduced any errors into their analysis. (I presume the correct information is in the Table and not in the 'site' column of the source data, because Balonghin is the only site that didn't select samples based on microscopic gametocytes and hence has a much lower density distribution.)

There is a third even lower female gametocyte density point in Mali (in spite of microscopically observed gametocytemia) that is truncated from Figure 2B.

Bobo-Dioulasso has a very high male fraction, very low measured female density, in spite of all points being microscopically detected and infecting large fractions of mosquitoes.

Figure 2B tells one narrative (threshold male gametocyte density below which transmission is less efficient). But the attached colors points by male gametocyte fraction, which I think can be thought of as correcting for imprecise female gametocyte density quantification when comparing the infectiousness vs. density curves.

Reviewer #2:

This paper follows from a previous *eLife* paper which quantified the relationship between female gametocyte densities and infectiousness to mosquitoes. This paper includes new estimates of male gametocyte densities which, in view of the changing sex-ratio at low female densities, provides a more complete description of the relationship and removes the need for the double hump previously seen and instead uses a simpler saturating relationship. It is not in itself an advance on statistical methods or the laboratory methods (described in another paper) and the sex-ratio was already thought to be important, but it is a step forward. In addition, there is a finding that the proportion of mosquitoes infected plateaus, but number of oocysts continues to rise with increasing gametocyte densities.

It is not clear how the density of male gametocytes or the sex-ratio were included in the model for the relationship between female gametocytes and the proportion of mosquitos infected. The methods merely states "Bayesian Markov Chain Monte Carlo techniques were used to fit the relationship and compare models as described previously". The Results section states "The best fit model shows negative density dependence". This model equation and parameter estimates may be in an Appendix but I could not find it. (The equation for gametocyte density and oocyst density is given in the legend to Figure 3).

In some cases, the fitted lines seem to miss the weight of the data (Figure 2A and C, and Figure 3B). The fitted lines are also dependent on some highly influential points at the righthand side of the graph.

Cameroon has few observations (n=13). Given that there is a skew to low gametocyte density infections, it is not obvious that the sample size is large enough to fit a line meaningfully.

In the Discussion section, the difference in the shape of the relationship in this compared to the previous paper is assumed to be due to more accurate automated methods and the inclusion of male gametocytes. As far as I can gather, different datasets were used here compared to the previous paper, so it could be a factor related to the datasets as well. It would be useful to know how much of the change in the shape of the relationship was due to the male gametocytes, and this could be investigated by using the previous model on the current datasets.

The relationship between gametocytes and infectivity are different between the datasets from Mali, Yaoundé and Burkina Faso. There is a brief mention in the Discussion section, but the reasons are not really known, and it should be stated that there is still considerable uncertainty, even with male and female gametocyte densities.

[Editors’ note: what now follows is the decision letter after the authors submitted for further consideration.]

Thank you for resubmitting your work entitled "Predicting the likelihood and intensity of mosquito infection from sex specific *Plasmodium falciparum* gametocyte density" for further consideration at *eLife*. Your revised article has been favorably evaluated by Prabhat Jha (Senior editor), a Reviewing editor, and three reviewers.

The manuscript has been improved but there are some remaining issues that need to be addressed before acceptance, as outlined below:

The reviewers agreed that this study represents an important body of work of a high general interest and a significant advance on the previously published paper. The current submission is much improved over the original version. Nonetheless, there are several issues that collectively, the reviewers thought still need to be addressed.

1) It has been proposed that the authors expand either the Introduction or the Discussion section by describing the previously fitted relationship and explicitly contrasting it with the new relationship. This would help to highlight the differences in the conclusions made in each of the two studies.

2) It would helpful if the authors would include some information explaining the importance of male densities in the context of the evolutionary theory of sex ratios. There is a reasonable body of theory and data (from *P. mexicanum* and *P. chabaudi* mostly) that can give ecological and evolutionary context to the results. Specifically, the fact that at low densities of gametocytes, males are limiting is something that has been called "fertility insurance". (there are papers by Gardner et al., 2003 and Ramiro et al, 2011 that explain and explore fertility insurance).

3) There should also be a more coherent integration of the oocyst results with the overall message of the paper. Total gametocytes are indicator to be the most important predictor for oocysts, but for prevalence it is female gametocytes with sex-ratio.

4) For completeness, the relationship between male-only gametocyte densities and infectiousness should be investigated to confirm that it is the weakest.

Motivated by the suspicion that some important data might be discarded by ignoring the microscopy densities, the authors should also indicate the performance of including both PCR and microscopy in the same model. Suggested fits were: gametocyte density by microscopy, geometric mean of male +female by PCR, geometric mean of female PCR + microscopy.

Amongst the specific comments, the following should be addressed.

Please mention and reference Da et al., 2015) that *P. falciparum* oocyst rise has also been observed.

Introduction, please drop "for the first time" from the text, as it has not: https://www.sciencedirect.com/science/article/pii/S0014489414002896#f0015 (Figure 1A.)

"Gametocyte density estimates by microscopy and qRT-PCR were correlated in Oueleesebougou (r=0.74), Bobo Dioulasso (r=0.27), Balonghin (r=0.56) and Yaoundé (r=0.91)." Correlation is not binary. The wording should reflect the wide range of correlation coefficients.

Subsection “Gametocyte sex ratios in natural infections”: Is the decreasing male-female ratio with increasing density still significant in a hold-one-out sensitivity analysis of the 4 sites? For example, what if Bobo is excluded?

Figure 1A: There are measurements with lower measured densities that do not appear on this figure.

Subsection “Infectivity in relation to gametocyte density”: It looks to my eyes that 200/μL is more like 30% infected.

Discussion section: I appreciate the inclusion of a previous comment in the sentence, "concurrent quantification […] leads to an improved estimation of total gametocyte biomass". However, the following sentence builds a false straw-man argument around the situation where that improved quantification would have to take the form of naively adding together the two imprecise measurements. As a rebuttal, take the following example:

Figure 1—figure supplement 1 in Mali shows that a qRT-PCR density of 50/μL is observed for microscopy measurements of between 20 and 300/μL. The Mali points in Figure 1 show that if the true female density is around 20/μL, then our expected male measurement would be about 2/μL, while if the true female density is 300/μL, the expected male measurement would be about 20/μL.

The point is that if we observe a (female+male) density of (50+2)/μL or (50+20)/μL, we should not expect a model that treats those as 52 or 70/μL total biomass to behave very well under the logarithmic measurement uncertainty. Rather, the previous paragraph would suggest that (50+2) is more consistent with 22/μL while (50+20) is more consistent with 320/μL!

As such, I am convinced the authors have done an important and more predictive measurement, but I disagree with the sentence "this work provides direct evidence of the epidemiological importance of male gametocytes" if that is intended to imply predominantly male-limited mating success (Discussion section).

Figure 1—figure supplement 1: Unless I am mistaken, all measurements are microscopy positive (except in Balonghin), so there should be 3 points missing from panel A (off the bottom?), 3 points from panel B (off the right?), 1 point from panel C (off the bottom?) and about 30 points from panel D, which are submicroscopic, but in the spirit of transparency might be presented also on a broken axis at x=zero.

Figure 1—figure supplement 1: The legend has the name Yaoundé in the wrong place. Move to after (C).

Density, rather than number of female, male parasites, is the technically correct term to be used.

Introduction: It's not clear to that separately quantifying the sexes is needed for estimating total gametocyte density. Is this because each sex may express different levels of the target gene?

Reconciling Figure 1A and 1B may be confusing because Figure 1A suggests the more gametocytes there are (because females make up the bulk of gametocytes), the more males there are. Some comments on why the pattern in Figure1B emerges would help.

A comment in the Discussion section on the relative ease of quantifying females vs total gametocytes or males would be useful information for others to take this forward.

Given their importance to this paper, the sex ratios could be mentioned in the text, currently they are in Figure 1 only.

---

## [Author Response]

[Editors’ note: the author responses to the first round of peer review follow.]

1) The new laboratory methods (quantification of male gametocytes and greater precision for female gametocytes) are described in other publications, and only very briefly here. Please expand on this topic in the discussion.

As requested, we have provided additional information on the assay methodology in the methods section and provide details on primer sequences in Table 2. We have also expanded the Discussion section to describe in more detail how male and female gametocytes were quantified.

2) Even with the final model including male gametocyte densities, there is uncertainty. There are substantial between-site differences, and within sites there is unexplained variation with more than a few points lying away from the fitted lines. Please provide explanations.

We have expanded our discussion of these important elements. It is important to realise that the actual fit is better than illustrated by Figure 2A. The line in Figure 2A presents the best fit, considering both female gametocyte densities and average male gametocyte density (for a given female density, based on the relationship displayed in Figure 1A), whilst the *x*-axis displays female gametocyte densities only. The full model fit to data is not possible to fully present in two dimensions but is displayed in the (3 dimensional) contour plot in Figure 2D. We display the simplified representation of the model in Figure 2A for presentation purposes and to enable direct comparison with the main figure of the original 2013 paper. Comparison with the original paper demonstrates two things: (1) the fit is considerably better and (2) the modelled relationship is much simpler, without the double plateau of the original paper. Nevertheless, there is still some difference between observed and predicted oocyst prevalence. Some degree of variability is unavoidable with biological experiments and this is especially the case with the membrane feeding assay which involves multiple organisms and complicated procedures.

As the reviewers note there is considerable between site variability. This is now better explained in the accompanying text and presented in Figure 2—figure supplement 1. In addition, 3 plausible biological mechanisms may explain the imperfect association: (1) Naturally acquired gametocyte antibodies can reduce transmission. We and others have demonstrated in recent manuscripts that naturally acquired antibody responses to gametocyte antigens can reduce both the prevalence of infected mosquitoes and the intensity of infection in these mosquitoes. Seasonal patterns in these responses might, as one reviewer points out, explain some of the variation between sites. (2) There may be a variation in the susceptibility of the mosquito strains to certain parasite genotypes. The importance of this phenomenon for transmission efficiency is currently unclear; the presented model may aid future studies into this topic. (3) Our panel of Pfs25 and PfMGET may not fully capture the biology of gametocyte maturation. *P. falciparum* gametocytes require 2-3 days of maturation upon release in the bloodstream from sequestration in the bone marrow and a reduced infectivity of older gametocytes has been hypothesized previously based on observations that gametocytes detected in the dry season (Ouedraogo, 2016) and late during infections (Johnston, 2012) may result in lower infection rates. We have addressed these uncertainties in the Discussion section of the revised manuscript.

3) There are a number of issues with the data analysis (correct data, sample size in one site, highly influential points, investigation into the Balonghin data). Please, re-analyze the results and provide a significantly revised interpretation.

We have confirmed that the data used for the analyses was correct. The error was in the preparing the data for submission when the labels for two of the sites were transposed. We are grateful to the reviewer for spotting this and it has been corrected for the resubmission.

We have noted the small sample size in the Cameroon data in the discussion and warned about over interpreting the differences between sites that may come from small sample sizes. However, we stress that the model was fit to the set of 148 data points as a whole and not to individual sites.

We do not believe that the fit of our model comes from a small number of highly influential points.

The Balonghin data are important to our analysis since this site includes many gametocyte densities that are undetectable by microscopy. We agree that they are not easily appreciated in the original figures due to the low gametocyte densities recorded at that site. To improve on this, we now show the data separately for each of the 4 sites (Figure 2—figure supplement 1).

Our responses to all the reviewers’ points about data analysis are detailed below.

4) It is stated that there is a change in the shape of the relationship between gametocyte densities and infectivity by including male gametocyte densities. However, the model may have greater prediction accuracy since two methods to estimate parts of the gametocyte population are used rather than one (the male and female gametocyte densities are added together in the model).

This is an important point for the overall message of the manuscript and we have expanded the Discussion section to highlight this. We acknowledge that this is a possibility although our analyses suggest that sex ratio is more relevant than total gametocyte biomass. Our analyses indicate that the greatest prediction accuracy comes from a model where transmission is dependent on female gametocyte density with a restriction in infection rates imposed at lower male gametocyte densities. Importantly, that model, despite an extra parameter, achieved a better fit (DIC 451.5) than a model based on simply calculating total gametocyte density (male + female gametocyte density estimates) that achieved a DIC of 501.7. Similarly, the model with total gametocyte biomass was inferior to the one where transmission was dependent on female gametocyte density alone (DIC = 481.3).

We have better explained this in the Discussion section of the revised manuscript.

In addition, the shape of the relationship does not appear to differ with the current datasets if only female gametocytes are used, compared to male and female together (the best fit for both would be the power function). It may be that there is a difference between the previous data and the current datasets.

To aid the readers understanding of the populations that contributed samples for the current manuscript we have expanded Table 1. We have also further described the study population in the methods and highlighted the points raised in the Discussion section.

The current data are indeed different from previous data though we do not feel that this alone explains the difference in results. Data in the Churcher et al. manuscript were from a cross-sectional survey of malaria infected children from Burkina Faso (highly similar in terms of study population to our four datasets) and a Kenyan population of children after antimalarial treatment. The association between gametocyte density and mosquito infection rates may be altered following malaria treatment (Dicko et al., 2016) due to an early sterilizing activity of circulating gametocytes that are still detectable by RNA transcripts. As a consequence, the Kenyan population in the original Churcher et al. manuscript is indeed different from our study populations and perhaps less useful to examine the association between gametocyte densities and mosquito infection rates (as most people will not be infecting mosquitoes immediately after drug treatment). However, the original Churcher et al. manuscript explicitly examined the shape of the association in the two datasets it used and both two populations gave near identical model fits. That finding (identical fit for samples collected post treatment and pre-treatment in our previous manuscript), the strong resemblance of 80/171 of the study population in Churcher, 2013 to our 148 study participants, and the consistent shape of the association between gametocyte density and mosquito infection rates at the four sites in the current manuscript make it highly unlikely that a difference in study population explains any of the differences between the original analysis and the current analysis.

Instead we believe that the difference between the current data and the previous data is primarily due to more accurate quantification of gametocyte densities. Gametocyte density estimates are considerably more precise in the current manuscript as QT-PCR is generally more precise than QT-NASBA and because the previous manuscript had a calibration curve dataset with an unknown male:female ratio. The high measurement error of the individual point estimates in the previous study is evident from the wide confidence intervals. We believe that this measurement error causes the proportion of infected mosquitoes to initially rising very rapidly and is responsible for the first of the previously observed double plateaus. Unfortunately, this cannot be tested retrospectively as blood samples from the original study are unavailable for re-analysis. Measurement error in the current study is much lower. This results in the simpler (and more parsimonious) function which rises more gradually. The reviewers are correct in that the power function fits best when either female alone or female and malegametocyte density is used. Therefore, it seems likely that the change in the shape of the best fit relationship is driven by improved gametocyte quantification and not the addition of information on the numbers of males (though variable sex ratio would have contributed to some of the measurement error in the previous manuscript). This is an important point and we have further stressed this in the Discussion section.

Reviewer #1:The subject of human-to-mosquito infectiousness and its determination by gametocyte density and other factors is a very important one. The authors make an important contribution here that is of general interest.I have three core concerns with the paper that should be addressed:1) The paper claims that we can now "reliably predict" mosquito infectiousness by including male and female gametocyte density quantification. However, the significant site-specific differences and other sources of unexplained variance are barely discussed.

In the original submission we aimed to stay below 1500 words for the total manuscript length which necessitated keeping the discussion to a minimum. We highly appreciate the points raised by the reviewer and agree that a greater explanation was required. In response to these comments and the feedback provided by the editor, we have considerably expanded our discussion. We have discussed possible reasons for imperfect prediction in greater depth and provided references for possible effects of the following factors: naturally acquired transmission reducing immunity (a likely contributor to model imprecision that may also have resulted in variation between sites); the use of antimalarial drugs (an unlikely factor since all experiments were conducted prior to antimalarials provided by the study team); genetic variation in the susceptibility of mosquito colonies to parasite isolates; and possible gametocyte fitness markers that are not captured by Pfs25 and PfMGET and technical issues related to RNA stability.

- Given the role that seasonality and immunity play on the underlying density distributions (Ouedraogo, 2015, Gerardin, 2015), more information in the Materials and methods section would be appreciated on year and season of sampling (e.g. January 2013 to November 2014 enrollment), enrollment protocol (e.g. microscopy positive gametocytes), and immunity (e.g. incidence per year) beyond "see Dicko, 2016" (not open access).

We have now provided full details of the sampled populations in Table 1. This includes the time of sampling in relation to the seasonality of malaria transmission and an estimate of malaria endemicity where possible (from surveys conducted within 3 years of the sampling points).

- You should discuss the implications of the Bobo, Cameroon, and Mali data consisting of microscopy-positive gametocyte carriers, but the Balonghin site only requiring Pfs25 positive.

We agree with the reviewers that this is important and have addressed this remark in the Results section and Materials and methods section. All sites recruited parasite carriers prior to antimalarial treatment. We believe the addition of the Balonghin site is a considerable strength since it gives information on lower gametocyte densities that can only be detected and quantified by molecular methodologies but can contribute considerably to transmission.

- You should include this information in Table 1, so the reader isn't forced to discover for herself the reasons for the very different gametocyte density ranges in spite of more similar asexual density ranges.

We have provided this information, on both enrolment criteria and time of data collection, in Table 1 and in detail in the dataset that is available online.

- Why is infectivity 3.65 times higher in Bobo-Dioulasso? The list of possible differences put forth in the Discussion section, "vector permissiveness, local parasite strains, human [immunity]" is vague and only serves to propagate the general confusion in the community. Without a satisfactory understanding, the claim of "reliably predicting" infectiousness is not very strong.

We appreciate this point and have considerably expanded our discussion, providing details on 4 possible explanations (immunity, vector permissiveness for parasite genotypes, gametocyte fitness and technical aspects related to RNA integrity) and 1 unlikely explanation (antimalarial drug use). We have discussed at length whether technical aspects related to RNA integrity (freeze-thaws in Bobo Dioulasso) could have contributed to our findings but there are no indications of this. We analysed the associations between microscopy estimated gametocyte densities and qRT-PCR estimated total gametocyte density for the 4 sites separately (Figure 1—figure supplement 1). Although qRT-PCR gametocyte density estimates are lower than microscopy estimates for a number of samples from Bobo Dioulasso, the majority of samples show good agreement between the two estimates and we do not consider this sufficient evidence to conclude RNA loss for that site and to remove the Bobo Dioulasso data from the current manuscript.

We would thus favour an approach where we present all data and explain in detail what possible elements could have explained outliers or site-to-site variation.

We have removed ‘reliably’ from statements on our ability to predict infection probability. Ultimately, membrane feeding assays are variable and hard to completely standardise between sites. However, we are still of the opinion that our manuscript provides a considerable improvement over previous efforts to understand the association between gametocyte density and mosquito infection rates.

2) This paper claims to make an important breakthrough in improved precision on female gametocyte density quantification. The Discussion section touches on this, but the implications of this are important and should be expanded upon.

We have incorporated this request and have better explained how our automated extraction and qRT-PCR (as compared to QT-NASBA) will have improved assay precision.

- Presently, there is insufficient detail on the determination of confidence intervals on density quantification. The only hint is the Materials and methods section sentence that they are "based on plate-specific calibration trendlines".

We have used the same statistical methods as the original 2013 paper. To aid the reader we have recapitulated them in the current submission.

- An example of where I doubt the veracity of the error quantification is in Figure 2B, where there are two samples with very low density, in spite of both being microscopically positive. Note the microscopy detection threshold is almost 2 orders of magnitude higher, or ~10-σ if error bars are 95% CI. Were these samples extreme outliers in leukocyte counts and the microscopy threshold is an invalid comparison?

We have addressed uncertainties in the estimated associations between gametocyte density and mosquito infection rates in more detail above. Regarding these two points, we do not have white blood cell counts from the slides and the slides were unavailable to address this question during our manuscript revision. One possible explanation may be RNA degradation in a subset of samples. There were no recorded technical issues during the sample collection in the field (e.g. long power outages or freezer problems that could have caused freeze-thaws that can be detrimental to RNA integrity), sample transport (World Courier shipment with temperature log confirming temperatures below -70C throughout the transport process) or upon receipt of samples in the laboratory at Radboud UMC, Nijmegen. However, this is a possible explanation that we have now specifically mentioned in the revised Discussion section.

Ultimately, estimation of gametocyte density is a stochastic process, which is determined by whether or not you get gametocytes in your sample by chance. Given that microscopy examines a substantially lower quantity of blood it appears likely that the density in these samples is likely to be over-estimated by the microscopy estimate (by chance a quantity of blood viewed under the microscope has more gametocytes than the average for the whole blood-source). The consistency between the gametocyte density by PCR and the% mosquito infected we believe supports this view.

- Does the female gametocyte density measurement error have significantly wider non-Gaussian tails?

PCR measurement error for the individual data points is quantified by estimating how far individual PCR CT values fall away from the true value (the CT trendline generated by serial dilution of high intensity sample). There is no evidence to suggest that this uncertainty does not follow a Gaussian distribution, and this was verified visually during the fitting process. Note that this measurement error is estimated on a logarithmic scale (as it is a ten-fold dilution series) so confidence interval estimates are non-symmetric in the Figures. We provide a reference for the full details of these methods in the manuscript (Walker et al., 2015).

3) A core assertion in this paper is that the addition of male gametocyte densities improves prediction. This could be caused by a combination of the following two factors: (1) mechanistically there is a threshold below which there are not enough male gametocytes present in the blood meal; or (2) the combination of two independent imprecise density measures gives a more accurate estimate of the "true" total gametocyte densities and hence more predictable infectiousness.The authors are strongly suggesting the first mechanism. However, there are indications that the latter mechanism is also playing an important role – take for example the Mali points that populate the lower-right quadrants of Figure 1A and 2A.

This is an important point and one we would like to thank the reviewers for raising. We have now specifically discussed the possibility that our improved predictions reflect an improved ability to quantify total gametocyte biomass. Nevertheless, we consider this unlikely to fully explain our findings as the best prediction accuracy came from the model with female density and a restriction in infection rates if male density was unfavourable (DIC= 451.5), which was considerably better than a model that simply calculated total gametocyte density by adding up female and male gametocyte density estimates (DIC = 501.7). Please see point 4 above for a further explanation. Regarding the points on the lower right quadrant of Figure 2A, these points (at higher female gametocyte densities) are still relatively far away from the best fit line if the model was dependent on the total sum of gametocytes (males + females). Given the skewed sex ratio at high female gametocyte densities the addition of males does little to increase total gametocyte density.

- A version of Figure 2B colored according to male gametocyte ratio would help discern how important each effect might be.

We agree that this will be of interest to the readers. This has been included as Figure 2—figure supplement 2.

The pattern is consistent with our best fit model: At low and intermediate female densities (between 1 and 100 female gametocytes per microlitre), a high proportion of males results in higher transmission because a minimum number of males are required for successful transmission. But for higher female densities, only a small proportion of males is needed to bring the absolute number of males over the threshold for successful transmission, and beyond that extra females contribute more to infectivity.

- A direct look at the Balonghin data, which in most plots is hiding in the corner of linear plots, would be instructive as it also includes submicroscopic gametocyte infections.

We agree that the Balonghin data, which we believe are highly relevant for the current manuscript since this site includes many gametocyte densities that are undetectable by microscopy, are not easily appreciated in the original figures. To improve on this, we now show the data separately for each of the 4 sites (Figure 2—figure supplement 1).

The source data has Balonghin and Bobo swapped compared to what appears in Table 1. The authors should verify this hasn't introduced any errors into their analysis. (I presume the correct information is in the Table and not in the 'site' column of the source data, because Balonghin is the only site that didn't select samples based on microscopic gametocytes and hence has a much lower density distribution.)

We apologise for this error. The reviewer is correct that the data presented in the paper are correct and the error was in the preparation of source data file sent for review. There were no errors in the analysis as a result of this and we have now corrected the error in source data.

There is a third even lower female gametocyte density point in Mali (in spite of microscopically observed gametocytemia) that is truncated from Figure 2B. Bobo-Dioulasso has a very high male fraction, very low measured female density, in spite of all points being microscopically detected and infecting large fractions of mosquitoes.

We thank the reviewer for spotting this. We have extended the axis to include this point. We consider it as likely that microscopy may have identified sparse gametocytes despite a low (i.e. ‘submicroscopic’) gametocyte density than imprecision of our molecular assays.

The low estimated gametocyte densities in several datasets were highlighted as a point that requires further discussion by both reviewers. Any density estimate comes with uncertainties. For microscopy, density estimates are influenced by white blood cell counts (against which gametocytes are enumerated) that can vary considerably between individuals and lead to imprecise estimates of gametocyte density per microliter. There is also a stochastic element in the detection of sparse gametocytes by microscopy. Potential clustering of gametocytes in the peripheral blood (Gaillard and Parasitol, 2003) may further contribute to imprecise gametocyte density estimates by microscopy.

This is illustrated by our attempt to re-read slides from the Mali study in preparation of the original manuscript describing the gametocyte sex ratio qRT-PCRs (Stone et al., 2017). For several slides, gametocytes were undetectable despite screening 200 microscopic fields by expert microscopists.

In addition to these shortcomings of microscopy, qRT-PCR has limitations. One possible explanation, albeit theoretical, is that RNA integrity was less well preserved for Bobo Dioulasso and that this disproportionally affected female transcripts (thus resulting in lower female gametocyte densities and more male-biased sex ratios). As indicated above, we have no evidence for problems in maintaining low temperatures in Bobo Dioulasso or elsewhere. When assessing the association between microscopy and qRT-PCR gametocyte density estimates (Figure 1—figure supplement 1), we also observed no evidence for RNA loss in this site. Inspired by the comments of the reviewers, we are currently testing mRNA integrity during freeze thaws in different RNA preservation buffers. These experiments may be of value for other groups aiming to quantify gametocyte sex ratio in field samples. However, none of the conclusions of the current manuscript would be affected, except for the considerable inter-site variation. As part of the current re-submission, we re-ran some of our key analyses on the dataset after excluding the Bobo Dioulasso data. None of the results were affected. For example, the relationship between gametocyte density and sex ratio remained apparent and significant (p-value from the Kruskal-Wallis test 0.0001) if only Mali, Cameroon and Balonghin data were included (see Author Response Image 1).

Figure 2B tells one narrative (threshold male gametocyte density below which transmission is less efficient). But the attached colors points by male gametocyte fraction, which I think can be thought of as correcting for imprecise female gametocyte density quantification when comparing the infectiousness vs. density curves.

We agree that this point should be further discussed and refer the reviewer to response (4) of replies to the Editor.

Reviewer #2:[…] It is not clear how the density of male gametocytes or the sex-ratio were included in the model for the relationship between female gametocytes and the proportion of mosquitos infected. The methods merely states "Bayesian Markov Chain Monte Carlo techniques were used to fit the relationship and compare models as described previously". The Results section states "The best fit model shows negative density dependence". This model equation and parameter estimates may be in an Appendix but I could not find it. (The equation for gametocyte density and oocyst density is given in the legend to Figure 3).

We have now made it clearer that this paper is linked to the previous one and that we used the same statistical methods. Based on a comment of reviewer 1, we have briefly summarized the statistical methodology that we used (and is provided in more detail in the previous manuscript).

We apologise for not including the appendix in the original submission. This is now included in Figure 2—source data 2 which describes the fitted models in full.

In some cases, the fitted lines seem to miss the weight of the data (Figure 2A and C, and Figure 3B). The fitted lines are also dependent on some highly influential points at the righthand side of the graph.

This is an important comment on highly influential points and we have spent considerable time in making sure our lines fit the data well and do not miss the weight of the data. To illustrate this, we have added additional figures.

For Figure 2A and Figure 3B, perhaps this is clearer now that we have added Figures Supplements which break the data down by site. Also, for Figure 2A it must be borne in mind that the line is the predicted infectivity based on female density and average male density at a given female density. The actual fit of the model (assessed by DIC) is better than it appears in the figure because it’s based on the actual male density, not just the one predicted by the female density.

We can see how the original data presentation may lead to the conclusion that the models miss the weight of the data in Figure 2C at first glance; but this is because of the many points in the bottom left of the graph (low male gametocytes and no infectivity) which can be missed.

Importantly, we do not share the concerns about the influential points on the righthand side of the graphs. If there was only one point on the righthand side of the graph then we would agree; but there are, for example, 3 points on the righthand side of Figure 2A and the line goes in-between them. Again, this is clearer from Figure 2—figure supplement 1, Figure 2—figure supplement 2 and Figure 3—figure supplement 1 which break the data down by site and demonstrate that the shape of the lines does not come from a small number of outliers.

Cameroon has few observations (n=13). Given that there is a skew to low gametocyte density infections, it is not obvious that the sample size is large enough to fit a line meaningfully.

We appreciate the point on the limited sample size. We made it clearer in our revised manuscript that all analyses were done on the complete dataset of 148 subjects and the limited number of observations for individual sites (notably Cameroon but also Bobo Dioulasso) warns against over-interpreting differences between sites. The main strength of the multi-site analysis is that there are no indications that the shape of the association between gametocyte density and mosquito infection prevalence or intensity is different between sites. We have specifically mentioned the limited sample size for 2/4 sites in the revised Discussion section.

In the Discussion section, the difference in the shape of the relationship in this compared to the previous paper is assumed to be due to more accurate automated methods and the inclusion of male gametocytes. As far as I can gather, different datasets were used here compared to the previous paper, so it could be a factor related to the datasets as well.

This is a fair point, but we consider it highly unlikely that the better predictions in the current study are affected by the study population. Please see the response to Editors comment 4 for a full explanation. To aid the reader in understanding the relevant characteristics of the populations who contributed samples for the current manuscript we have nevertheless clarified the study population more clearly in the Materials and methods section and Discussion section and included further explanation in the Discussion section.

It would be useful to know how much of the change in the shape of the relationship was due to the male gametocytes, and this could be investigated by using the previous model on the current datasets.

This is, obviously, one of the key messages of the manuscript. Two major improvements have been implemented since the original paper. First, we performed automated extraction of RNA, switched to qRT-PCR and based our estimates on pure male and female trendlines. The latter is potentially relevant since sex ratio fluctuations in vitro can fluctuate and trendlines with variable proportions of males and females may thus lead to different gametocyte estimates in the original approach where mixed sex trendlines were used to translate Pfs25 mRNA signal to gametocyte density.

Secondly, we incorporated sex ratio in the models. These improvements are now explained in more detail in the Discussion section.

The first set of improvements is purely related to laboratory improvements and not directly related to the incorporation of sex ratio in the statistical methodology. It therefore allows for an assessment of the direct merits of incorporating sex ratio in the equations, with the caveat that the individual procedural improvements cannot be disentangled (e.g. we did not run mixed sex trendlines on all plates for qRT-PCR). We can, however, run the original model for Pfs25 gametocyte density (as in Churcher, 2013 but with automated extraction, qRT-PCR and single-sex trendlines), total gametocyte density and female density with incorporation of sex ratio. The original model did not fit as well as the model presented in Figure 2A (DIC = 557). The details are described in Figure 2—source data 2. Further explanation is also given in response to Editor 4 where we highlight that we think it is the improved precision of gametocyte density which causes the shape change, not the presence of male parasites.

The relationship between gametocytes and infectivity are different between the datasets from Mali, Yaoundé and Burkina Faso. There is a brief mention in the Discussion section, but the reasons are not really known, and it should be stated that there is still considerable uncertainty, even with male and female gametocyte densities.

This is a crucial point that reviewer 1 also raised. We have expanded the Discussion section on these.

[Editors' note: the author responses to the re-review follow.]

1) It has been proposed that the authors expand either the Introduction or the Discussion section by describing the previously fitted relationship and explicitly contrasting it with the new relationship. This would help to highlight the differences in the conclusions made in each of the two studies.

We agree and have contrasted the new and previous relationships and discussed the consequences in the Discussion section:

‘The analyses show that the relationship between the density of gametocytes and transmission can be adequately described by a simple saturating relationship, and that a more complicated curve with two inflection points (which generates a double plateau) does not improve model fit [1]. […] The new work indicates that even gametocytocidal drugs with incomplete efficacy would reduce transmission.’

2) It would helpful if the authors would include some information explaining the importance of male densities in the context of the evolutionary theory of sex ratios. There is a reasonable body of theory and data (from P. mexicanum and P. chabaudi mostly) that can give ecological and evolutionary context to the results. Specifically, the fact that at low densities of gametocytes, males are limiting is something that has been called "fertility insurance". (there are papers by Gardner et al., 2003 and Ramiro et a., 2011 that explain and explore fertility insurance).

We have improved our Introduction and Discussion section and incorporate these two references in our discussion of the previous work on this topic.

Introduction: ‘Gametocyte sex-ratio in *P. falciparum* is typically female biased but may differ between infections (7,8) and during infections (7,9), and may be influenced by immune responses that differentially affect male and female gametocytes (10). Fertility assurance explains why under certain conditions, such as a low gametocyte density, gametocyte sex-ratio may become less female biased to ensure all female gametes are fertilized (11)’

Discussion section: ‘The work confirmed previous evidence that the proportion of male gametocytes is negatively associated with total gametocyte density (5,8,26), which may reflect a strategic investment in male gametocytes to maximize the likelihood of transmission in low-density infections (fertility assurance) (5,27)’

3) There should also be a more coherent integration of the oocyst results with the overall message of the paper. Total gametocytes are indicator to be the most important predictor for oocysts, but for prevalence it is female gametocytes with sex-ratio.

We go into more detail on the fit of the oocyst model and contrast it to the infectivity model in the revised Results section:

‘Total gametocyte density (sum of females and males) was able to predict mean oocyst density better than female gametocyte density alone. The higher complexity of the density model (requiring the fitting of the negative binomial distribution) meant that it was not possible to fit a model analogous to the best fit model on infectivity (i.e. a function dependent on female gametocyte density but with reduced transmission at low male densities). Care should be taken interpreting the relative importance of females and male gametocytes in predicting oocyst density as this result should be validated with a larger dataset.’

And also, in the Discussion section:

‘There were insufficient data to fit a model where oocyst density was dependent on female density but with a penalty for low male densities which was the best fit model for infectivity. Whether such a model would fit the oocyst density data better than the model presented here is a question for future research’

4) For completeness, the relationship between male-only gametocyte densities and infectiousness should be investigated to confirm that it is the weakest.

We have added this to Figure 2—source data 2 and male density is indeed the weakest and commented on this in the revised Results section:

‘Poorer statistical fits were observed for models where the proportion of mosquitoes with oocysts was described by either the density of females alone (DIC=481.3), the density of males (DIC = 538.8), or total gametocyte density (sum of male and female gametocytes, DIC=501.7).’

Motivated by the suspicion that some important data might be discarded by ignoring the microscopy densities, the authors should also indicate the performance of including both PCR and microscopy in the same model. Suggested fits were: gametocyte density by microscopy, geometric mean of male +female by PCR, geometric mean of female PCR + microscopy.Amongst the specific comments, the following should be addressed.

We have added this analysis to Figure 2—source data 2 and referenced this in the Results section of the main text:

“Other models, including both qRT-PCR and microscopy, showed poorer performance in predicting mosquito infection prevalence (Figure 2–source data 2).”

We preferred to use the arithmetic mean rather than the geometric mean because many samples from Balonghin have microscopy 0, and geometric means are always 0 if one component is 0.

Please mention and reference Da et al., 2015) that P. falciparum oocyst rise has also been observed.Introduction, please drop "for the first time" from the text, as it has not: https://www.sciencedirect.com/science/article/pii/S0014489414002896#f0015 (Figure 1A.)

We have adjusted this as suggested and changed the Introduction.

‘The relationship between gametocyte density and oocyst intensity is also investigated as this may show a different relationship with gametocyte density [13] and recent work has highlighted the importance of parasite intensity in mosquito-to-human transmission [14]. Both the number of infected mosquitoes and their parasite load may need to be considered when assessing the human reservoir of infection’.

"Gametocyte density estimates by microscopy and qRT-PCR were correlated in Oueleesebougou (r=0.74), Bobo Dioulasso (r=0.27), Balonghin (r=0.56) and Yaoundé (r=0.91)." Correlation is not binary. The wording should reflect the wide range of correlation coefficients.

We have changed the wording accordingly in the Results section:

“Gametocyte density estimates by microscopy and qRT-PCR were positively correlated in all sites, but there was variation in the strength of correlation: Ouelessebougou (r=0.74), Bobo Dioulasso (r=0.27), Balonghin (r=0.56) and Yaoundé (r=0.91) (Figure 1—figure supplement 1).”

Subsection “Gametocyte sex ratios in natural infections”: Is the decreasing male-female ratio with increasing density still significant in a hold-one-out sensitivity analysis of the 4 sites? For example, what if Bobo is excluded?

The relationship still holds if Bobo is left out. It is also significant if either Balonghin or Yaoundé is left out. Mali represents almost half the data and the relationship is no longer significant if Mali is removed. We also mentioned this in the Results section:

“This association remained apparent and statistically significant if any of the study sites with a smaller number of observations was removed (i.e. with removal of data from Bobo Dioulasso, Balonghin or Yaoundé).”

Figure 1A: There are measurements with lower measured densities that do not appear on this figure.

We apologise for this error and have corrected it in the revised submission.

Subsection “Infectivity in relation to gametocyte density”: It looks to my eyes that 200/μL is more like 30% infected.

The reviewer is correct, and we have corrected it to say approximately 30%. The actual figure predicted by the curve is 32.4%.

Discussion section: I appreciate the inclusion of a previous comment in the sentence, "concurrent quantification… leads to an improved estimation of total gametocyte biomass". However, the following sentence builds a false straw-man argument around the situation where that improved quantification would have to take the form of naively adding together the two imprecise measurements. As a rebuttal, take the following example:Figure 1—figure supplement 1 in Mali shows that a qRT-PCR density of 50/μL is observed for microscopy measurements of between 20 and 300/μL. The Mali points in Figure 1 show that if the true female density is around 20/μL, then our expected male measurement would be about 2/μL, while if the true female density is 300/μL, the expected male measurement would be about 20/μL.The point is that if we observe a (female+male) density of (50+2)/μL or (50+20)/μL, we should not expect a model that treats those as 52 or 70/μL total biomass to behave very well under the logarithmic measurement uncertainty. Rather, the previous paragraph would suggest that (50+2) is more consistent with 22/μL while (50+20) is more consistent with 320/μL!As such, I am convinced the authors have done an important and more predictive measurement, but I disagree with the sentence "this work provides direct evidence of the epidemiological importance of male gametocytes" if that is intended to imply predominantly male-limited mating success (Discussion section).

We apologise to the reviewer for building a false straw-man, obfuscation did not drive this, but rather we did not fully understand the point the reviewer made. We agree with the reviewer that simply adding two imprecise point estimates together may not necessarily improve model fit even if oocyst prevalence is determined by total gametocyte density if there is high measurement error. The Bayesian methods we employ takes this into account to some extent as each total gametocyte density estimate in the posterior distribution is generated by randomly sampling from distributions of female and male gametocyte density estimates which vary reflecting the measurement error as assessed by the PCR calibration trend lines. We agree with the reviewer that adding imprecise gametocyte density estimates together which have non-normally distributed measurement error may also generate bias, particularly when data is left censored (as is necessary to generate meaningful prevalence estimates). To reflect this, we have rephrased this section and now no longer suggest that our study provides direct evidence for the epidemiological importance of male gametocytes though we keep it as a hypothesis as we still feel that it is the most likely. Further studies are needed to directly confirm this.

‘The concurrent quantification of female and male gametocytes leads to an improved estimation of total gametocyte biomass and could explain why models with information on both sexes were more accurate. Similarly, combining data from microscopy and PCR could also improve overall precision when both techniques have relatively high measurement error. Though improved accuracy of gametocyte density though combining imprecise measures cannot be discounted the model fit to total gametocyte density (male plus female) gave a poorer fit than the model where transmission was dependent on female density unless there were very low male parasite densities. This suggests that the number of male gametocytes may become a limiting factor for transmission at low gametocyte densities although future studies that directly relate infectivity to mosquitoes to female and male gametocyte densities are needed to confirm this association.’

Figure 1—figure supplement 1: Unless I am mistaken, all measurements are microscopy positive (except in Balonghin), so there should be 3 points missing from panel A (off the bottom?), 3 points from panel B (off the right?), 1 point from panel C (off the bottom?) and about 30 points from panel D, which are submicroscopic, but in the spirit of transparency might be presented also on a broken axis at x=zero.

The reviewer is correct, and we have included all submicroscopic points in the revised Figure 1—figure supplement 1.

Figure 1—figure supplement 1: The legend has the name Yaoundé in the wrong place. Move to after (C).

We have corrected this.

Density, rather than number of female, male parasites, is the technically correct term to be used.

We agree with the reviewers and have corrected this throughout the manuscript.

Introduction: It's not clear to that separately quantifying the sexes is needed for estimating total gametocyte density. Is this because each sex may express different levels of the target gene?

This is a valid point. The target transcripts are enriched in male and female gametocytes but not exclusively expressed by either gender. We have chosen an approach where male and female gametocytes are quantified separately based on trendlines of male and female gametocytes. An alternative approach, which we now specifically mention in the Discussion section, would be an approach where gametocytes are quantified based on a target transcript that is equally expressed by male and female gametocytes.

‘[…] may benefit from an assay that targets a transcript with identical expression levels in male and female gametocytes [24] or a multiplex male-female gametocyte assay that avoids uncertainties in quantifying total gametocyte biomass from two separate assays.’

Reconciling Figure 1A and 1B may be confusing because Figure 1A suggests the more gametocytes there are (because females make up the bulk of gametocytes), the more males there are. Some comments on why the pattern in Figure1B emerges would help.

We have added an explanation in the Results section:

‘The fact that the relationship is linear on the log scale means that the percentage of gametocytes that were male decreased with increasing total gametocyte densities (Figure 1B, p < 0.001 [Kruskal Wallis test]).’

A comment in the Discussion section on the relative ease of quantifying females vs total gametocytes or males would be useful information for others to take this forward.

We have addressed this, as indicated above

‘[…] may benefit from an assay that targets a transcript with identical expression levels in male and female gametocytes [24] or a multiplex male-female gametocyte assay that avoids uncertainties in quantifying total gametocyte biomass from two separate assays.’

Given their importance to this paper, the sex ratios could be mentioned in the text, currently they are in Figure 1 only.

We have added sex ratios to Table 1 and the following line in the Results section:

“The median proportion of gametocytes that was male was 25% (interquartile range 13% – 39%) (Table 1, Figure 1).”